# Development of New Dermato-Cosmetic Therapeutic Formulas with Extracts of *Vinca minor* L. Plants from the Dobrogea Region

**DOI:** 10.3390/ijms242216234

**Published:** 2023-11-12

**Authors:** Ana-Maria Neculai, Gabriela Stanciu, Anca Cristina Lepădatu, Luiza-Madălina Cima, Magdalena Mititelu, Sorinel Marius Neacșu

**Affiliations:** 1Faculty of Medicine, Department of Biochemistry, Ovidius University of Constanta, 900470 Constanta, Romania; anamneculai89@gmail.com; 2Department of Chemistry and Chemical Engineering, Ovidius University of Constanta, 900527 Constanta, Romania; 3Department of Natural Sciences, Faculty of Natural and Agricultural Sciences, Ovidius University of Constanta, 900470 Constanta, Romania; anca.cristina.lepadatu@univ-ovidius.ro; 4Faculty of Pharmacy, Department of Pharmaceutical Chemistry, Titu Maiorescu University of Bucharest, 020956 Bucharest, Romania; luiza.cima@prof.utm.ro; 5Clinical Laboratory and Food Hygiene Department, Faculty of Pharmacy, Carol Davila University of Medicine and Pharmacy, 020956 Bucharest, Romania; 6Faculty of Pharmacy, Department of Pharmaceutical Technology and Bio-Pharmacy, Carol Davila University of Medicine and Pharmacy, 020956 Bucharest, Romania; neacsusorinelmarius@gmail.com

**Keywords:** indole alkaloids, vincamine, semisolid pharmaceutical preparations, rheological profile, antioxidant capacity, antimicrobial activity, inflammation

## Abstract

A new trend in the use of indole alkaloids from natural products is the preparation of topical pharmaceutical formulations with applications in the field of regenerative medicine. These formulations can be characterized through the ease of administration, the proven healing action of indole alkaloids, the protection of skin lesions, and the assurance of oxygen permeability. Based on the numerous benefits that indole compounds extracted from the *Vinca minor* plant show externally, the purpose of this study was to develop new semi-solid biocomposites for topical application obtained from hydroalcoholic macerates of 40%, 70%, and 96% concentrations from the stems and leaves of the *Vinca minor* L. plant from the Dobrogea area. A total of 12 pharmaceutical formulations (named P1–P12) were prepared for which the physicochemical properties, pH, thermal stability, spreading capacity, and rheological behavior were determined. The optimal formulas with antioxidant and antimicrobial capacity were evaluated and determined (P3, P4, P9, and P10). Antioxidant activity was elicited using the photochemiluminescence method. The microorganisms used for the evaluation of antimicrobial activity were Gram-positive *Staphylococcus aureus* (ATCC 25923), Gram-negative *Escherichia coli* (ATCC 25922), and a fungal species, *Candida albicans* (ATCC 900288). The study of the rheological profile for the obtained composites revealed Newtonian, pseudoplastic, and thixotropic fluid behaviors. Following determinations using the photochemiluminescence method, the best antioxidant activity was obtained in the P3 and P9 preparations. The results of the antimicrobial analysis confirmed that both the leaves and the stems of the *Vinca minor* plant represent a valuable source of antibacterial substances, and the biocomposites analyzed may represent an alternative in the realization of new pharmaceutical preparations with topical applications based on hydroalcoholic macerates obtained from the *Vinca minor* plant.

## 1. Introduction

*Vinca minor* belongs to the Apocynaceae family and is the most characteristic medicinal plant [1,2]. Native to northern Spain and western France, *Vinca minor* has naturalized in several areas, including the Dobrogea area of Romania. In traditional medicine, *Vinca minor* is well known for its sedative properties, its ability to lower blood pressure, and its healing potential [3].

Pharmacologically, the most valuable indole alkaloid extracted from the *Vinca minor* plant is vincamine. Vincamine improves brain metabolism, protects neurons, improves peripheral blood flow, increases the supply of oxygen to the nerve cells, and is used for the prevention and treatment of cerebrovascular insufficiencies and disorders through increasing cerebral blood flow [4,5]. Externally, it exhibits wound-healing effects through speeding up the healing process of lesions [3].

The use of indole alkaloids derived from natural sources has been a recent trend in the development of new topical pharmaceutical formulations with potential applications in regenerative medicine [4]. It was found that the associations among different substances, which present a therapeutic action directed to the same area of interest, has demonstrated a better efficiency for their use in different pharmaceutical formulations [6,7]. In addition, numerous studies provide information on the antibacterial [8,9,10,11] and antifungal [12,13] activities of bioactive compounds in hydroalcoholic macerates obtained from the plant *Vinca minor* L. The natural antimicrobial capacity of plant extracts was used to preserve processed foods and to manufacture pharmaceutical and dermato-cosmetic products, in alternative medicine and herbal therapies [14].

Recent findings indicate that natural phenolic compounds present in this plant have an antibiotic therapeutic effect on biofilm formation [15]. Based on research, it has been established that these polyphenolic compounds can interfere with the creation of bacterial biofilm, preventing its formation, as bacteria have complex systems of regulating certain metabolic activities depending on population density [16]. Testing the antimicrobial activity of semi-solid preparations is an important alternative strategy in the control of infectious diseases caused by antibiotic-resistant pathogens [17,18].

For the characterization of a semi-solid biocomposite, the rheological profile is a quality attribute of the preparation. For the rheological study of semi-solid biocomposites, it is important to know the rheological parameters in order to characterize the rheological profile, which provides important information about some final physicochemical properties of pharmaceutical forms, such as stability, but also to establish some technological parameters used in pre-formulation and formulation [19,20].

Since there are no data in the literature on the incorporation of indole compounds in pharmaceutical forms for topical administration, this study aims to valorize the *Vinca minor* plant from the Dobrogea area for therapeutic purposes, to develop original dermal preparations with pharmaceutical potential containing hydroalcoholic macerates obtained from the leaves and stems of the *Vinca minor* plant. Thus, 12 new biocomposites (P1–P12) were designed and prepared for topical administration using hydroalcoholic macerates of 40%, 70%, and 96% concentration, obtained from the leaves and stems of the *Vinca minor* plant in combination with other substances of plant origin with antiseptic, antimicrobial, and regenerative action to enhance the therapeutic effect. The 12 dermal formulations (P1–P12) were analyzed in regard to physicochemical, organoleptic, and rheological properties, and the composites considered as having the optimal formulation (P3, P4, P9, and P10) were studied for antioxidant and antimicrobial activity. The P3, P4, P9, and P10 formulations contain hydroalcoholic macerates of 70% concentration, which is the solvent with the highest extraction power, according to our previous studies [21]. To evaluate the stability of the 12 biocomposites, pharmacotechnical characterization was carried out to analyze the physicochemical properties, pH determination, thermal stability, viscosity analysis, and spreadability. Antioxidant activity was determined via the photochemiluminescence method and antimicrobial activity was tested using the diffusimetric method. Antimicrobial activity was tested against two categories of pathogenic microorganisms, bacteria and fungi, using the following reference strains: Gram-positive (*Staphylococcus aureus* ATCC 25923) and Gram-negative (*Escherichia coli* ATCC 25922) pathogenic bacterial strains and pathogenic fungal strains (*Candida albicans* ATCC 10231).

These microbial species were selected for their presence, as commensal strains, in the normal microbiota of the skin and mucous membranes, but which, under certain conditions of imbalance, following associated diseases (immunocompromising, old age, etc.), can become pathogenic. *Staphylococcus aureus* is a Gram-positive bacterium that causes suppurative infections or septicemia, i.e., skin infections, leading to secondary wound infection [22]. *Escherichia coli* is a Gram-negative bacterium, commensal in the digestive tracts of mammals and humans, frequently detected in a wide variety of food sources: water, raw meat, vegetables, and dairy products. It is an important indicator of sanitation, with its presence on the skin indicating fecal contamination of an animal and/or human nature [23]. *Candida albicans* is an opportunistic fungal organism found in the normal microbiota of the skin, gut, and genitourinary tract, but can become a pathogen under conditions of low immunity or due to an imbalance in bacterial microflora [24].

The objective of these studies is to evaluate the possibilities of incorporation of the *Vinca minor* plant harvested from the Dobrogea area into different pharmaceutical formulations, especially for potential use in dermal preparations with therapeutic properties.

## 2. Results

### 2.1. Characteristics of Vinca minor Extract-Based-Biocomposites

The results obtained following the pharmacotechnical analysis of the 12 biocomposites are presented in Table 1 and Table 2. According to the results obtained, the prepared formulations are stable and homogeneous.

### 2.2. Rheological Behavior of Biocomposites

The rheological characteristics of the semi-solid composites encoded in Table 1 and Table 2 are presented in Table 3 and Table 4.

The increase in shear stress with shear speed indicates a non-Newtonian behavior of the 12 analyzed biocomposites. Figure 1 shows the flow curves for the 12 preparations analyzed. 

Figure 2 illustrates the rheograms obtained for the P1–P12 formulations analysed.

Figure 3 shows the linearization of the rheological parameters for the 12 composites analyzed. The correlation coefficient of the linearization obtained for the 12 preparations ranged between 0.9953 and 0.9992 (Table 4).

Among the 12 biocomposites, the spreading capacities of each preparation tested showed good consistency, the differences being explained by their different compositions. The rheological profile indicates an unstable elasticity at high shearing forces for some formulations, so that, above the flow threshold, some of them lose their consistency, passing into the field of plastic deformations. The ones that show good stability and a return to the initial shape even after applying shear forces above the yield point are formulations P3, P4, P9, and P10. The spreading capacity results for the four dermal formulations considered optimal (P3, P4, P9, and P10) are shown in Figure 4.

A small variation is observed for the spreading capacity of the tested compositions after 30 days of preparation, confirming the quality of the dermal formulations.

### 2.3. Determination of Antioxidant Activity for Vinca minor Extract-Based Biocomposites

The total antioxidant capacity of the encoded biocomposites P3, P4, P9, and P10, compared to Trolox^®^ standard, was quantified according to the ACL (Analytik Jena AG, Thuringia, Germany) procedure for the stock solution and for the dilution of the stock solution 1:100. The quantitative results, expressed in mg TE/100 g sample, are shown in Table 5.

### 2.4. Determination of Antimicrobial Activity for Vinca minor Extract-Based Biocomposites: Statistical Data Processing for Comparative Study of the Antimicrobial Effect of P3, P4, P9, and P10 Composites

The analyzed formulations contain 70% hydroalcoholic macerate of *Vinca minor* plant stem for P3 and P4 and 70% hydroalcoholic macerate of *Vinca minor* plant leaf for P9 and P10, with addition of zinc oxide for P4 and P10 and silver sulfadiazine for all four proposed formulations. Results from in our previous study [25] regarding *Vinca minor* 70% hydroethanolic macerate antimicrobial activity displayed close values for both extracts prepared from the leaf and from the stem of *Vinca minor* plant, meaning that phytoconstituents with similar antimicrobial activity are present in extracts obtained from both studied plant materials. Due to the fact that, in the initially performed assays [24], equivalent volumes of negative control extraction solvent (70% ethanol) exercised antimicrobial effects against the same tested strains, ethanol was evaporated from all experimental variants using a rotary evaporator at a temperature of 20 °C. Thus, we can appreciate that the antimicrobial effect was exercised by the bioactive phyto-compounds and by added synthetic ingredients and not by ethanol.

The results (Figure 5, Figure 6 and Figure 7) showed good antimicrobial activity for the four biocomposites, P3, P4, P9, and P10, against the reference strains *S. aureus*, *E. coli*, and *C. albicans*. Increased antimicrobial activity was observed for preparations P4 and P10, which additionally contain zinc oxide, for all tested microbial strains.

The one-way ANOVA test (Figure 8) was applied to compare the diameters of the inhibition zones between the P3, P4, P9, and P10 biocomposites in the same well diameter; this was performed for the three experimental variants (9 mm, 11 mm, and 15 mm). All extracts reacted in a dose-dependent manner: the larger the diameter of the well (understanding increasing quantities and concentrations of biocomposites), the larger the diameter of the inhibition zone around the disposal area; there were significant differences between samples for all microbial tested strains.

One-way ANOVA test applied to compare the diameters of the inhibition zones of the P3, P4, P9, and P10 variants with the positive control (C): hydroalcoholic macerate 70%, at the same well diameter, with calculation of the F-factor for all three microbial strains tested (F = 8460.055 for testing on the Gram-positive reference strain *S. aureus* ATCC 25923; F = 4461.055 for Gram-negative reference strain *E. coli* ATCC 25922; F = 4461.055; F = 2546.193 for the fungal reference strain *C. albicans* ATCC 10231) showed significant differences (*** *p* < 0.001) compared to the positive control for all experimental variants tested (Figure 9, Figure 10 and Figure 11).

A Tamhane post hoc multiple comparation test (test of homogeneity of variances based on mean: Leven statistics = 14.805, *p* < 0.001) revealed that the mean inhibition zone diameter (mm) was statistically significantly different between all experimental variants taken two by two (*** *p* < 0.001) but not between P3 and P9 (*p* = 0.999) in the case of tests on Gram-positive *Staphylococcus aureus* ATCC 25923.

A Tamhane post hoc multiple comparation test (test of homogeneity of variances based on mean: Leven statistics = 8.792, *p* < 0.001) revealed that the mean inhibition zone diameter (mm) was statistically significantly different between all experimental variants taken two by two (*** *p < 0.001)* in case of tests on Gram-negative *Escherichia coli* ATCC 25922.

There was a statistically significant difference between groups as determined by Tamhane post hoc multiple comparation test (test of homogeneity of variances based on mean: Leven statistics = 15.251, *p* < 0.001) between C-P3, C-P4, C-P9, and C-P10 experimental variants (*p* < 0.001) but not between P3 and P9 (*p* = 0.996) or P4 and P10 (*p* = 0.999), in case of test on *Candida albicans* ATCC 10231.

### 2.5. Determination of Anti-Inflammatory Activity for Vinca minor Extracts Based Biocomposites

Recent studies have highlighted the antioxidant, antibacterial, and antitumor effect of the different extracts obtained from the different parts of some *Vinca* species; the effects are due to the phytonutrients in the composition of the plants [17]. We set out to analyze the anti-inflammatory potential of the biocomposites made with the extracts obtained from Vinca minor; as a benchmark, we used a pharmaceutical preparation for external use, authorized and recognized for a certain anti-inflammatory effect, with diclofenac as the active substance.

For the evaluation of the anti-inflammatory effect on experimentally induced inflammations in laboratory animals, only the P4 and P10 formulations were chosen, due to the fact that they also presented the highest antimicrobial activity. The experimental results of the evaluations of the anti-inflammatory effects carried out on the formulas P4 and P10, with compositions as presented in the data in Table 6 and Table 7, are presented in Figure 12, Figure 13, Figure 14 and Figure 15. The evolution of the edema-inhibition effect (I%) over time for the preparations used (P4, P10, and Diclofenac gel as reference) compared to the untreated control group are presented in Figure 13 and Figure 15.

Both preparations studied (P4 and P10) showed an obvious anti-inflammatory effect. For preparation P10, we found the most effective action on both types of induced inflammation. Figure 12 and Figure 14 show graphically the inflammatory level at different time intervals, starting 2 h after the induction of inflammation. In the case of their treatments with the tested preparations, a reduction in the inflammatory process was observed.

Regarding the percentage of inhibition of inflammation compared to the control group, the following results were determined: In the case of inflammation induced by kaolin 4 h after administration; the Diclofenac gel, chosen as reference, has an inhibition percentage of 70.19%; P4 has an inhibition percentage of 43.04%; P10 has an inhibition percentage of 55.44%. After 24 h, the highest percentage of reduction in the inflammatory process between the two formulations is presented by P10 with 56.84%, so a reduction of over 50%, with P4 there was 44.5% (Figure 13). In the case of dextran-induced edema, the strongest inflammation inhibitory effect of the two formulations tested was also presented by P10 with a maximum after 60 min. (51.22%); P4 showed a reduction of 44.72% and the reference preparation (Diclofenac gel) showed a reduction of 66.26% (Figure 15).

According to the experimental data, a reduction in inflammations of slightly over 50% is observed in the case of the P10 formula; thus, this formula has moderate anti-inflammatory potential, which cannot be neglected and which brings advantages in the therapeutic use of the preparation.

## 3. Discussion

The development of the 12 biocomposites (P1–P12) for topical administration was based on the numerous benefits that the indole compounds extracted from the *Vinca minor* plant demonstrated for dermal application [26,27,28]. The composition of these preparations was formulated to ensure biocompatibility, adhesion, and a possible improved therapeutic effect. The formulations were prepared in the form of L/H or H/L emulsions, containing hydroalcoholic macerates obtained from the leaves and stems of the *Vinca minor* plant as the main substances used for their healing, antimicrobial effect; this is associated with various substances that have their own pharmacological action. The 12 new pharmaceutical formulations developed in this study were differentiated by the different percentages of hydroalcoholic macerate (40%, 70%, and 96%) contained, and, respectively, other components with antiseptic and antimicrobial properties in the preparations with topical application. The optimal formulations (P3, P4, P9, and P10) were found to be those with Ag sulfadiazine and 70% hydroalcoholic macerate in their composition. Ag sulfadiazine is known for its antiseptic, antimicrobial, and microorganism growth-inhibitory properties [29]. Compounds P4 and P9 containing Zn oxide show better antimicrobial activity due to the antiseptic and antibacterial properties of Zn oxide.

For each preparation, the organoleptic properties, pH, thermal stability, spreadability and rheological profile were determined. Following the analysis of the organoleptic properties, the results confirmed that the 12 biocomposites were homogeneous and translucent, with a characteristic odor emitted by the components, these properties correspond to quality control. Each pH value test was performed in triplicate, and the values resulting from the tests were within the accepted limits set by F.R. X. (4.5–8.5) and were also similar to the physiological pH of the skin. As for the analysis of the spreadability study of the 12 pharmaceutical formulations prepared, the method prescribed by P. Ojeda–S. Arbussa was used. Based on the results, it was found that all 12 of the pharmaceutical formulations showed a good spreading capacity, influenced by the presence of hydroalcoholic macerates obtained from the leaf and stem of the *Vinca Minor* plant.

The rheological studies performed resulted in rheograms that highlighted that all P1–P12 formulations presented higher shear stresses at decreasing shear speeds than those measured at increasing shear speeds, thus recording hysteresis loops. Preparations P6 and P11 were an exception to this finding, as they showed increases in shear stresses with increasing shear speeds compared to decreasing speeds. This also explains the different alignment of the rheograms for these preparations containing hydroalcoholic macerates obtained in ethyl alcohol at a low concentration (40%). Also, the preparations containing hydroalcoholic macerates at a low concentration (40%) (P5, P6, P11, and P12) showed lower viscosity values than those containing hydroalcoholic macerates in 70% and 96% concentrations, respectively.

Composites formulated with hydroalcoholic macerates obtained from the leaf of the *Vinca minor* plant, P7–P12, showed larger ranges of values for all rheological parameters (viscosity, velocity gradient, and shear stress) than those obtained from the stem of the *Vinca minor* plant, P1–P6. This is due to the different compositions in different types of Vinca alkaloids obtained in hydroalcoholic macerates, in leaf versus stem, for all comparative ethanol concentrations [21].

The addition of ZnO in the formulation of both the hydroalcoholic leaf macerates and the *Vinca minor* plant stem preparations, obtained in low concentrations of ethyl alcohol (40%), did not influence the values obtained for apparent viscosity. The addition of ZnO to preparations of leaf and stem plant hydroalcoholic macerates in 70% and 96% ethanol increased the thixotropy of the preparations compared to those without ZnO.

Regarding the comparison of the results with a known rheological model, it is found that all P1–P12 preparations in the form of H/L or L/H emulsions follow the Ostwald de Waele model, with flow index n < 1, for all formulations. This confirms that the preparations exhibit pseudoplastic fluid profile.

From the data presented in Table 5, it can be seen that the four biocomposites analysed showed good antioxidant capacity. Biocomposites P3 and P9 that do not contain zinc oxide in their composition show a significantly higher antioxidant capacity than those which contain zinc oxide (P4 and P10). The stock solution for P3 and P9 shows inhibition, although the TEAC result does not match the standard curve; the 100-fold-diluted samples have good antioxidant capacity. The explanation is that the zinc oxide, present in the P4 and P10 biocomposites, adsorbs on its surface the bioactive compounds present in the biocomposite, causing their removal by filtration, in the sample preparation phase for the determination of antioxidant capacity.

Of considerable importance in the choice of microbial strains to be tested was the difference in the architecture of the bacterial envelope, which is complex in Gram-negative bacteria, with three principal layers: the outer membrane, the peptidoglycan (murein) cell wall, a complex molecule with a three-dimensional heteropolymer, composed of glycan and a peptide, and the cytoplasmic or inner membrane. Gram-positive bacteria have a simpler envelope and a thinner wall, with only two layers because the lack of outer membrane. The outer membrane plays a major role in protecting Gram-negative organisms from the environment by excluding toxic molecules and providing an additional stabilizing layer around the cell. Because the outer membrane indirectly helps stabilize the inner membrane, the peptidoglycan mesh surrounding Gram-negative cells is relatively thin. Gram-positive bacteria often live in harsh environments, but they lack a protective outer membrane. To withstand the turgor pressure exerted on the plasma membrane, Gram-positive microorganisms are surrounded by layers of peptidoglycan many times thicker than is found in Gram-positive bacteria [30].

In yeasts, the cell wall has a complex structure and has an important role in the relationship with the external environment or the substrate to which the fungus adheres and in the defense of the yeast cell. Thus, the wall is a complex structure, representing ~25% of the dry weight of yeast cells, in *Candida albicans* consisting of chitin, (1-3)-D-glucan, (1,6)-beta-glucans, lipids, and peptides, incorporated in a protein matrix [12,13]. The enzyme-resistant nature of chitin and chitosan make them ideal coatings for dormant forms of organisms (cysts), with fungi appearing to have unusual resistance relative to other phyla [12,13].

This architecture will influence the selective permeability of the microbial envelope and its synthesis mechanisms, with consequences in terms of sensitivity/resistance to different bioactive compounds with potential antibiotic effects [30].

We investigated the pragmatic and idiosyncratic behavior of some microbial strains (Gram-positive, Gram-negative, and fungal reference strains) in the presence of four originally formulated biocomposites. The comparative study of the microbial growth inhibitory effect displayed by biocomposites P3, P4, P9, and P10—containing 70% hydroalcoholic (the optimal extractive variant) macerates of *Vinca minor* leaves and stems, with Ag sulfadiazine present for all variants and zinc oxide present for P4 and P10—revealed statistically significant differences compared to the positive control (70% hydroalcoholic macerate), for all tested microbial strains. This means that the antimicrobial effect of the crude extract was enhanced by the addition of the synthetic components zinc oxide and Ag sulfadiazine with antimicrobial, biocompatible, biodegradable, and non-toxic properties [29]. The antimicrobial activity was species-dependent; the best inhibitory effect was against Gram-positive *E. coli* strains and *C. albicans* fungal strains; there was a lower inhibitory effect against *S. aureus* Gram-negative strains, depending on the complexity of the microbial cell wall architecture and on phytoconstituents and added synthetic components. Similar results were obtained with *V. minor* extracts against Gram-positive bacteria and a lower inhibitory capacity against the Gram-negative ones [29]. These inhibitory effects are due to phytoconstituents such as vincamine [31] and common plant flavonoids such as caffeic acids, rutin, quercitrin, isoquercitrin, quercetin, chlorogenic acid, etc. [32], that act synergistically. Another report proved that vincamine is effective against *E. coli* and *S. aureus* [31] and that the flavonoids quercetin, chlorogenic acid, rutin, and kaempferol were generally more effective against *E. coli* and *Pseudomonas aeruginosa* (Gram-negative) than against *Enterococcus faecalis* and *S. aureus* (Gram-positive) [32]. Other Vinca species extracts, such as *V. herbacea* extract containing elevated levels of rutin, have a good inhibitory effect on *S. aureus*, compared to *E. coli* [33], but also against *Aeromonas hydrophyla* [34].

Statistical tests and graphical representation of the mean values of the diameters of the inhibition zones for the four experimental variants P3, P4, P9, and P10, compared to the positive control variant (Figure 8, Figure 9, Figure 10 and Figure 11), displayed the efficiency of the experimental variants P4 and P10 and lower antimicrobial effect of P3 and P9 for all tested strains. The addition of ZnO to preparations of leaf (P4) and stem (P10) plant hydroalcoholic macerates in 70% ethanol increased the microbial inhibitory effect of the preparations compared to those without ZnO.

Also, the P4 and P10 formulations showed obvious anti-inflammatory effects on experimental models induced in laboratory animals.

Our results recommend the exploitation of the therapeutic potential of the *Vinca minor* plant in the pharmaceutical and phytotherapy industry, both in medicinal formulations for internal and external use.

## 4. Materials and Methods

### 4.1. Chemicals and Equipment

The substances used were purchased from Elemental Romania: lanolin, vaseline, cetyl alcohol, stearic acid, zinc oxide, shea butter, white wax, hyaluronic acid, vegetable collagen, and vitamin E. Ag sulfadizine was purchased from Sigma-Aldrich, Darmstadt, Germany, and fish collagen was extracted from Black Sea Nisetru, in the Chemistry–Physics Laboratory of Ovidius University Constanta, Faculty of Pharmacy.

Adequate equipment was used for the methods of physical–chemical analysis. The pH determination was performed using a Hanna Instruments (Woonsocket, United States) instruments multiparametric pH meter to measure pH values. The P. Ojeda–S. Arbussa [35] method was used for the spreadability study of the composites. The rheological profile was performed with the Rotational viscosimeter ST-2020 R, manufacturer Laboquimia, Spain, through which the rheological measurements were made to obtain the rheological parameters: viscosity, shear speed, rotation speed, and shear stress [36].

Photochem Equipment, Analytik Jena AG, Germany, was used to determine antioxidant activity via the photochemiluminescence method [37].

Microbial strains and growth conditions: Antimicrobial activity was tested against two categories of pathogenic microorganisms, bacteria and fungi, using the reference strains (Sanimed) Gram-positive *Staphylococcus aureus* ATCC 25923 and Gram-negative *Escherichia coli* ATCC 25922 pathogenic bacterial strains and pathogenic fungal strains *Candida albicans* ATCC 10231. Culture media was purchased from Sigma-Aldrich, Germany. *S. aureus* and *E. coli* were grown using Tryptone Soya Broth (TSB, 30 g/L)/Tryptone Soya Agar (TSA, 15 g/L) and *C. albicans* was grown using Sabouraud 2% Dextrose Broth (SDB, 30 g/L)/Sabouraud 4% Dextrose Agar (SDA, 65 g/L).

### 4.2. Preparation of Alcoholic Extracts from Vinca minor L.

Plant material, leaves, and stems of *Vinca minor* L. were collected during the flowering period in May 2022 from Dobrogea area, Romania (Figure 16). The dried plant material was ground using a laboratory mill into a fine powder.

The first phase in the development of semi-solid biocomposites for dermal application was the preparation of hydroalcoholic macerates, which consisted in the maceration of 50 g of dry and ground vegetable product obtained from the leaves and stems of the Vinca Minor plant, respectively, over which ethyl alcohol 40%, 70% was added, with 96% up to a 500 mL (ratio 1:10) for 10 days under optimal conditions, protected from light and humidity, in a cool place [20]. The liquid collected upon filtration was stored in sterile, dark containers.

### 4.3. Investigation Methods

#### 4.3.1. Formulation of Semisolid Biocomposites

The 12 semi-solid formulations based on hydroalcoholic macerates obtained from the leaves and stems of *Vinca minor* were prepared according to the composition shown in Table 6 and Table 7. The preparation process was similar and included the preparation of the two phases, the hydrophilic and lipophilic phases: Homogenization was performed in a water bath and successive cooling to 25 °C. In the hydrophilic phase, the hydroalcoholic macerates were added, where wetting and dispersion was carried out. Preparation of the lipophilic phase was performed by weighing and mixing the substances in containers of suitable capacity, solubilizing the components in a water bath up to a temperature of 70 °C, followed by shaking the mixture. The two phases were brought to the same temperature. Mixing of the lipophilic phase with the hydrophilic phase was performed by continuous shaking for 15 min, with slow cooling. The resulting semi-solid preparations were transferred to the corresponding storage container and analysed after 30 h.

After finishing the preparation process of the semi-solid biocomposites P1–P12, the samples of the obtained 12 preparations were analysed further for characterization: appearance analysis, pH determination, thermal stability, rheological profile, and spreadability.

Appearance was analysed by examining with a magnifying glass (4.5×) a sample of the sample stretched in a thin layer on a microscopic slide [38].

The pH determination (using Hanna instruments multiparametric pH meter) was performed on the aqueous phase after extraction of the samples with distilled water (1:10) and separating through heating in a water bath at 60 °C; then, homogenization was performed for 10 min. [39].

Determination of thermal stability was carried out by maintaining a sample of each preparation analysed at two temperature steps for 8 h: 2 °C and 40 °C. The samples were placed in weighing bottles with lids and the bottles were kept in the oven and in the refrigerator at the mentioned temperatures, after which the appearance of the samples was examined; the samples should be homogeneous [40].

The spreading capacity of the biocomposites was determined using the Ojeda Arbussa method [36] on the samples 30 h after preparation and after 30 days. The spreading diameter of 1 g of sample placed between two 20 × 20 cm glass plates for 1 min was evaluated. The mass of the upper plate was standardized to 125 g, and on top of the plate, different weights were gradually added (50 g, 100 g, 150 g, 200 g, 250 g, 500 g, and 750 g) at intervals of 1 min; then, the spreading zones reached by the each sample were calculated. The results were expressed in terms of the surface area of the spread as a function of the applied mass, according to Equation (1):S_i_ = d_i_^2^ (π/4)(1)
where S_i_ – spreading area (mm^2^) resulting after the applied mass “i” (g); d_i_ – mean diameter (mm) reached by the sample.

#### 4.3.2. Rheological Measurements

For the rheological study of the semi-solid biocomposites (P1–P12) the cylindrical system method with known geometry was used. Following rheological measurements (viscosity measurements at different rotational speeds, increasing and decreasing), the shear speed and shear stress were calculated. The rheograms and flow curves were constructed for the obtained mean values of the rheological parameters. Readings of apparent viscosity η (cP) were performed in triplicate at increasing and decreasing rotational speeds ω (rpm).

The rheological parameters used for the rheological measurements were systematized in the mathematical equations presented in Table 3. The apparent viscosity, designated as the viscosity value at a specific shear rate, was determined using Equation (2) in Table 3. This allowed the construction of viscosity curves. Using Equation (3) in Table 3, rheograms were obtained for the biocomposites studied, considering the values of shear stress τ and shear speed D. The calculation of shear speed D (s^−1^) was performed using Equation (4) in Table 3, which was related to the values of rotational speed ω (rpm). In addition, the specific constant R was used for each axis of rotation to determine the shear speed D in relation to the chosen rotational speed ω (rpm). The calculation of the shear stress τ was determined by applying Equation (5):η = f(D)(2)
D = f(τ)(3)
D = ωxR(4)
τ = ηxD(5)
τ = kxD^n^(6)

The rheological study of the samples was evaluated by increasing and decreasing ω speeds in the range of 4–200 rpm. In the rheological analysis with the rotational viscosimeter ST-2020 R, manufacturer Laboquimia, Spain, the time interval between determinations and the time used for each measurement was 10 s. For viscosity measurements, R5 and R6 pivots were used, which were adequate for the viscosity values of the preparations. They were used for the calculation of the shear speed D in correlation with the rotational speed ω of the pivot.

Regarding the comparison of the results with a known rheological model (Ostwald de Waele), Equation (6) was applied.

#### 4.3.3. Determination of Antioxidant Activity

Determination of the total antioxidant capacity of fat-soluble substances (ACL) for preparations P3, P4, P9, and P10 was performed via the photochemiluminescence method [21,41]. For the analysis, 1 g of sample of each formulation was dissolved in 10 mL solvent (n-butyl alcohol). The solution obtained was then filtered through filter paper to remain clear. For each biocomposite studied, the antioxidant capacity was determined for the stock solution and for the 1:100 stock solution dilution with reagent R1 from the kit according to the ACL procedure (Analytik Jena AG). Aliquots of 5 μL were taken from each sample (from the supernatant) and exposed to external radiation produced by a phosphor-coated Hg lamp. The standard reagent kit (Analytik Jena, Germany) of the ACL procedure was used for the analyses: R1 (dilution solvent), R2 (buffer reagent), R3 (photosensitive reagent), and R4 (sized reagent). Mixtures were prepared according to Table 8.

The calibration curve (Figure 2) was calculated using a series of standard solutions containing 0.5, 1.0, 2.0, and 3.0 nM Trolox (6-hydroxy-2,5,7,8-tetramethylchroman-2-carboxylic acid), a vitamin E derivative. Determinations were performed in triplicate and were expressed as nM Trolox equivalents (TE)/µL sample and as an mg (TE)/100 g d.w. sample [42,43,44,45]. The correlation coefficient has a value of 0.9948.

#### 4.3.4. Determination of Antimicrobial Activity

The microbiological determinations were intended to test the antimicrobial potential of topically administered dermal preparations in different pharmaceutical formulation options (Figure 5, Figure 6, Figure 7, Figure 8, Figure 9, Figure 10 and Figure 11) incorporating hydroalcoholic macerates in 70% ethanol (the optimal extraction option) of *Vinca minor* leaves and stems, containing plant bioactive principles shown in our previous studies [25] to have antimicrobial activity. Antimicrobial activity was tested on two categories of pathogenic microorganisms, bacteria and fungus, using the following reference strains: Gram-positive *Staphylococcus aureus* ATCC 25923, Gram-negative *Escherichia coli* ATCC 25922 pathogenic bacterial strains, and pathogenic fungal strains *Candida albicans* ATCC 10231.

Qualitative evaluation of antimicrobial activity was performed using a version of the Kirby–Bauer diffusimetric method [45] which replaced classical antibiotics micro-tablets with different amounts of the tested pharmaceutical preparations. To determine the efficiency of the samples as microbial growth inhibitors, increasing amounts (meaning increasing concentrations) of these compounds were placed (respecting aseptic conditions) in wells of different diameters: Ø = 9 mm (0.15 g preparation), Ø = 11 mm (0.20 g preparation), and Ø = 15 mm (0.40 g preparation). The method was based on the diffusion mechanism of the four pharmaceutical formulation compounds through the culture medium inoculated with the mentioned bacterial and fungal test strains. A measure of 0.5 McFarland standardized turbidity inoculum was prepared through direct homogenization in sterile saline of three–five colonies from exponentially grown 18 h microbial culture plate. A PCA (Oxoid) Petri dish was inoculated with 1 mL of 0.5 McFarland bacterial suspension (1.5 × 10^8^ CFU/mL). After medium solidification, wells of different diameters were made and grammes of the pharmaceutical preparations to be tested were placed into each well. For each sample, plates were inoculated in triplicate. After an incubation period of 22 ± 2 h at 37 °C, the results reading and interpretation were performed qualitatively, according to NCCLS 2020 [46,47], through measuring (using a graduated ruler) and evaluating in mm the zones of microbial growth inhibition around the area of sample arrangement in the well. The presence of any zone of inhibition was interpreted as sensitivity (S) and its absence as resistance (R). Results were expressed as the mean of three measurements ± SD.

#### 4.3.5. Determination of Anti-Inflammatory Activity

The evaluation of the anti-inflammatory activity was determined on formulas P4 and P10—the formulas with the best antimicrobial activity. For this purpose, 8 groups of 10 Wistar rats were used.

The animals, weighing 210 ± 15 g, were initially kept at laboratory conditions for 2 days for accommodation (experimental room temperature of 22 ± 2 °C, humidity of 40–50%).

The study protocol was approved by the Scientific Research Ethics Commission of Carol Davila University of Medicine and Pharmacy, established under the Animal Protection (Code of Ethical Conduct 128/11.02.2023), Animal Welfare Act 1999 [48,49].

To evaluate the anti-inflammatory action, two inflammatory agents were used that were administered intra-plantar by injection of 0.1 mL of 10% kaolin suspension and 0.2 mL of 6% dextran solution [50].

One group constituted the control group; two groups were treated with the P4 and P10 formulas; and one group was treated with Diclofenac gel 1% produced by the Fiterman company. On the paw in which the edema was induced, the test preparation was applied uniformly in a thin layer of ~0.25 g gel.

Group 1—Control group, untreated;

Group 2—Reference group, treated with Diclofenac gel 1%;

Group 3—Tested group, treated with the P4 formula;

Group 4—Tested group, treated with the P10 formula.

The evaluation of the anti-inflammatory effects of the P4 and P10 formulas were compared to Diclofenac gel, which is a recognized anti-inflammatory preparation on the Romanian pharmaceutical market.

The determinations were made against control groups (untreated animals from control group).

The volume of the rat’s paw was measured using a plethysmometer, Ugo Basile 7140 (Gemonio, Italy). The measurements were performed at intervals of 2 h, 4 h, 6 h, and 24 h (for the edematous agent, 10% kaolin suspension) and at intervals of 30 min, 60 min, 90 min, and 120 min from the induction of edema (for the edematous agent, 6% dextran solution). 

The average value of anti-inflammatory edema (expressed in mL) and the percentage of the edema-inhibition effect for each batch were calculated according to the following formula:Edema-inhibition effect (I) % = (X control − X treatment agent/X control) × 100(7)
where X treatment agent represents the average value of the edema produced by the tested gel (Diclofenac gel or formula P4, respectively P10); X control represents the average value of the edema produced in the control group in the same time interval after the administration of the edematous agent.

All determinations were performed in triplicate, and the results were expressed as mean ± SD (standard deviation). Statistical evaluation of clinical results was performed using Student’s *t* test (*t* test) and analysis of variance (ANOVA) [51,52,53].

### 4.4. Statistical Data Processing

For data processing and comparison between the results of the antimicrobial activity analysis for the four semi-solid pharmaceutical formulations (P3, P4, P9, and P10) and the positive control (70% hydroalcoholic macerate), the one way ANOVA test in IBM SPSS software (version 29) was used.

## 5. Conclusions

New pharmaceutical formulations were developed (total of 12), in the form of H/L or L/H emulsions, using hydroalcoholic macerates with the following concentrations: 40%, 70%, and 96%. The formulations were developed from the leaves and stems of the *Vinca minor* plant. To enhance the healing and antimicrobial effect of the hydroalcoholic macerate from the dermal formulations, substances with their own pharmacological action were added. These included the following: zinc oxide, for its antifungal and antibacterial effects; Ag sulfadiazine, for its antiseptic effect; and hyaluronic acid, which is used for tissue regeneration.

The obtained results support the use of extracts from the aerial parts of *Vinca minor,* harvested from the Dobrogea area, in the treatment of skin lesions. Formulations P3 and P9, which showed good antioxidant capacities, can also be successfully applied in anti-aging therapies. Formulas P4 and P10 were found to have more pronounced antimicrobial action. These are recommended for applications in the pharmaceutical industry as alternative natural preservatives and disinfectants. The relative chemical compositions found in a given *Vinca minor* were found to be correlated with the observed antimicrobial potential of that specimen. The antimicrobial potential of the tested composites—P3, P4, P9, and P10—was microbial-species- and dose-dependent. Of course, the good antimicrobial effect that we observed was due to the synergistic action of a plethora of phytoconstituents. This was reflected in the observed antimicrobial activity against bacterial (Gram-positive and Gram-negative) and fungal strains; however, it is likely that the 70% ethanol-extracted bioactive compounds being combined with various substances that have their own well-known pharmacological activities could have enhanced the antimicrobial activity through a synergistic effect (zinc oxide for its antifungal and antibacterial effects; Ag sulfadiazine for its antiseptic effects).

The results of the study confirm that the proposed pharmaceutical formulations, based on hydroalcoholic macerates from the *Vinca minor* plant, represent an alternative to classic antibiotics in the development of new topical pharmaceutical preparations with disinfectant activity.

## Figures and Tables

**Figure 1 ijms-24-16234-f001:**
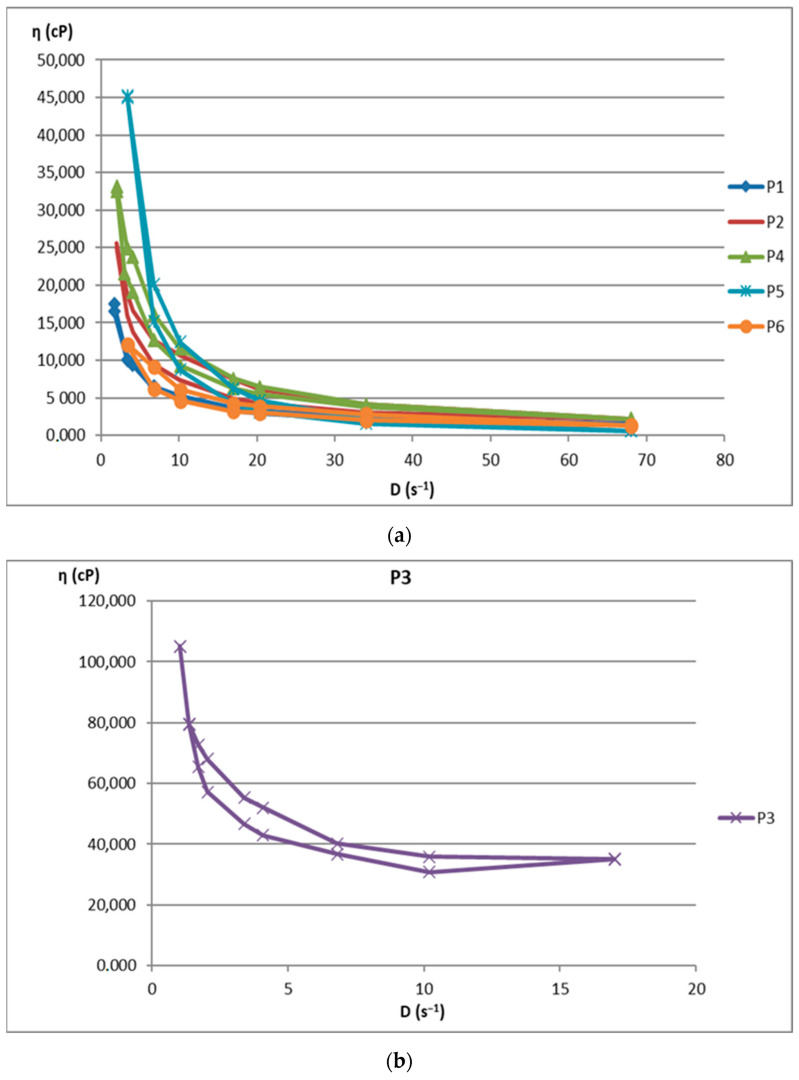
Flow curve for (**a**): P1–P6 biocomposites; (**b**): P3 biocomposite; (**c**): P7–P12 biocomposites.

**Figure 2 ijms-24-16234-f002:**
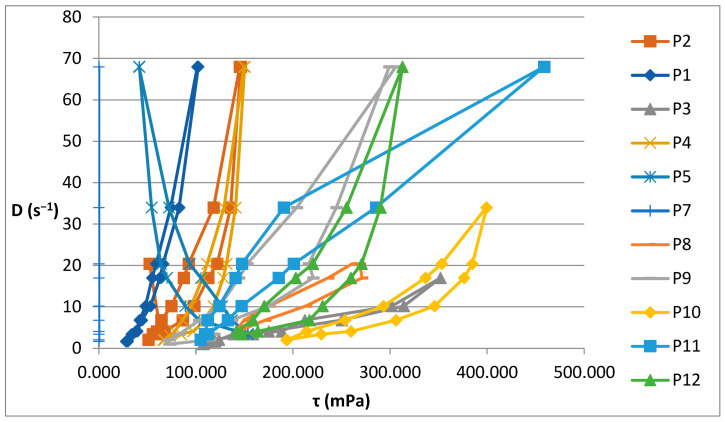
Rheograms for P1–P12 biocomposites.

**Figure 3 ijms-24-16234-f003:**
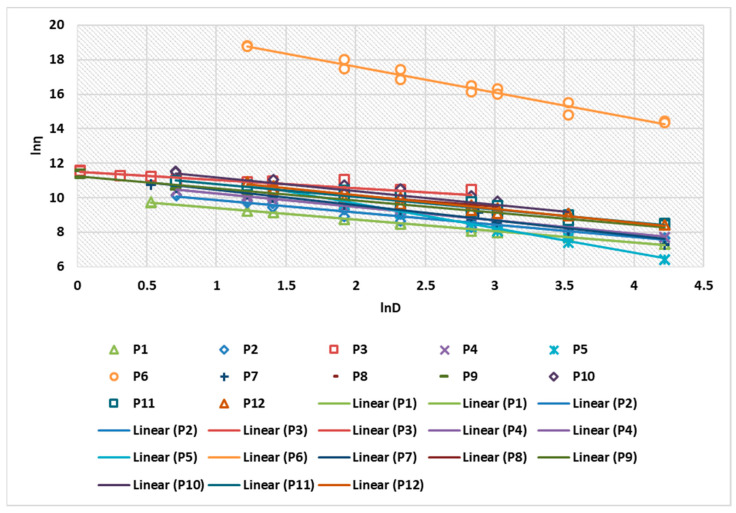
Linearization of rheological parameters for P1–P12 biocomposites.

**Figure 4 ijms-24-16234-f004:**
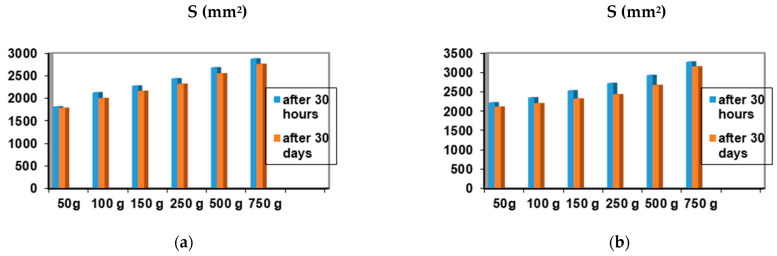
Spreadability of composites: (**a**) P3 after 30 h and after 30 days; (**b**) P4 after 30 h and 30 days; (**c**) P9 after 30 h and 30 days; (**d**) P10 after 30 h and after 30 days.

**Figure 5 ijms-24-16234-f005:**
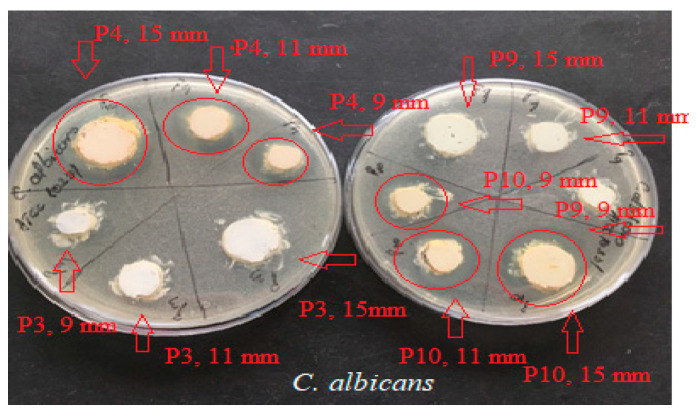
Antimicrobial activity of P3, P4, P9, and P10 for *C. Albicans* ATCC 10231. The red arrows mark the correspondence between the studied biocomposite and its disposition on the culture medium; the diameter of the placement well is mentioned. The red circle marks the appearance of a microbial growth inhibition zone around the biocomposite disposal well.

**Figure 6 ijms-24-16234-f006:**
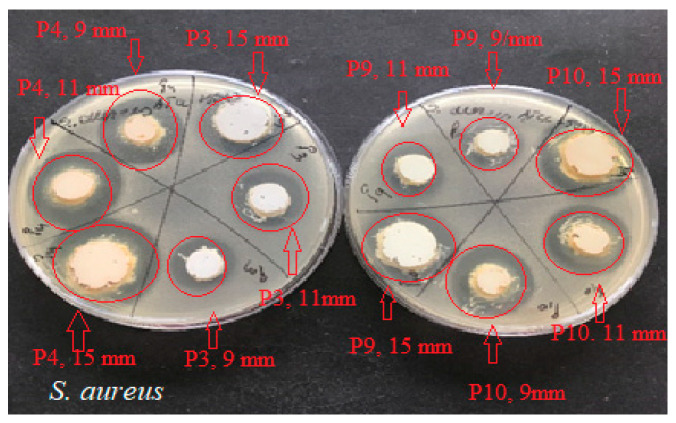
Antimicrobial activity of P3, P4, P9, and P10 for *S. aureus* ATCC 25923. The red arrows mark the correspondence between the studied biocomposite and its disposition on the culture medium; the diameter of the placement well is mentioned. The red circle marks the appearance of a microbial growth inhibition zone around the biocomposite disposal well.

**Figure 7 ijms-24-16234-f007:**
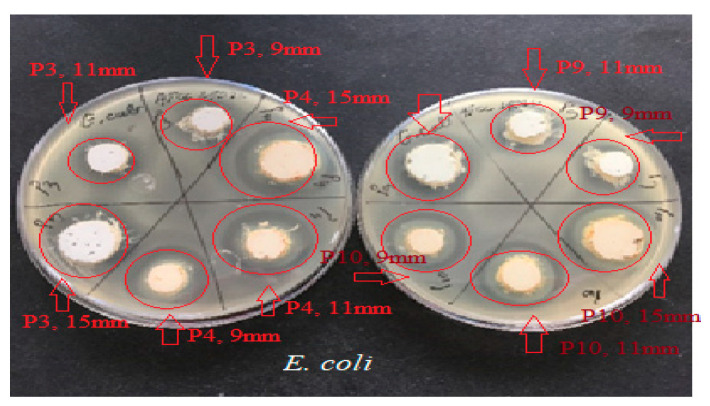
Antimicrobial activity of P3, P4, P9, and P10 preparations for *E. coli* ATCC 25922. The red arrows mark the correspondence between the studied biocomposite and its disposition on the culture medium; the diameter of the placement well is mentioned. The red circle marks the appearance of a microbial growth inhibition zone around the biocomposite disposal well.

**Figure 8 ijms-24-16234-f008:**
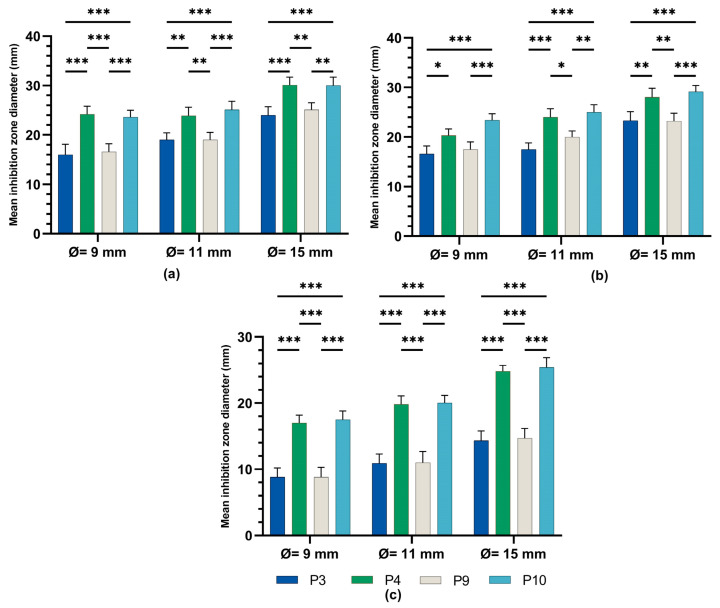
Inhibitory activity of composites P3, P4, P9, and P10 on (**a**) staphylococcus aureus, (**b**) *E. coli*, and (**c**) candida albicans ANOVA with Tamhane post hoc test (*** *p* < 0.001, ** *p* ≤ 0.005, * *p* < 0.05).

**Figure 9 ijms-24-16234-f009:**
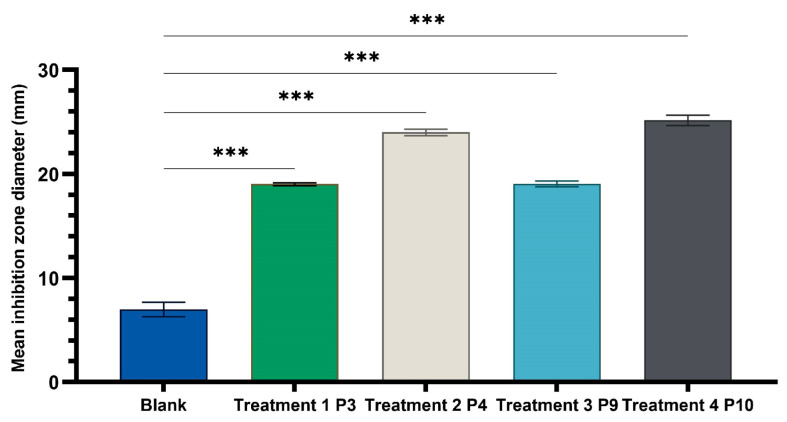
One-way ANOVA with Dunnett post hoc test result for comparison of experimental variants P3, P4, P9, and P10 with the positive control C (70% hydroalcoholic macerate), for testing on the Gram-positive reference strain *S. aureus* ATCC 25923 (*** *p* < 0.001).

**Figure 10 ijms-24-16234-f010:**
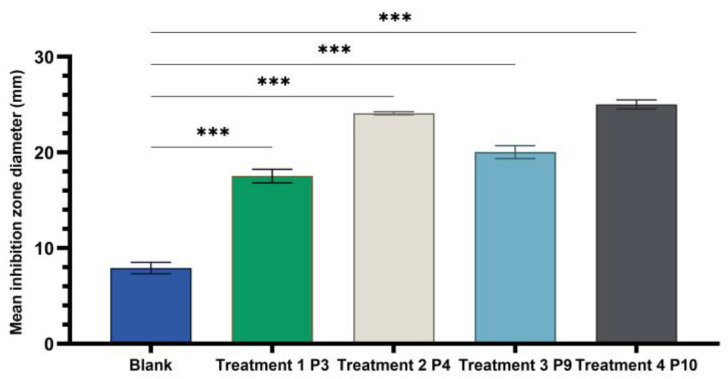
One-way ANOVA with Dunnett post hoc test result for comparison of experimental variants P3, P4, P9, and P10 with the positive control C (70% hydroalcoholic macerate), for testing on the Gram–negative reference strain *E. coli* ATCC25922 (*** *p* < 0.001).

**Figure 11 ijms-24-16234-f011:**
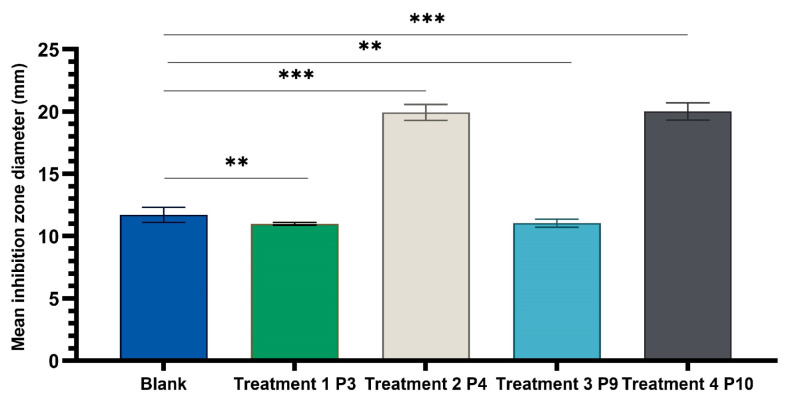
One-way ANOVA with Dunnett post hoc test of experimental variants P3, P4, P9, and P10 with the positive control variant (70% hydroalcoholic macerate), for testing on the reference fungal strain *C. albicans* ATCC 10231 (** *p* < 0.01, *** *p* < 0.001).

**Figure 12 ijms-24-16234-f012:**
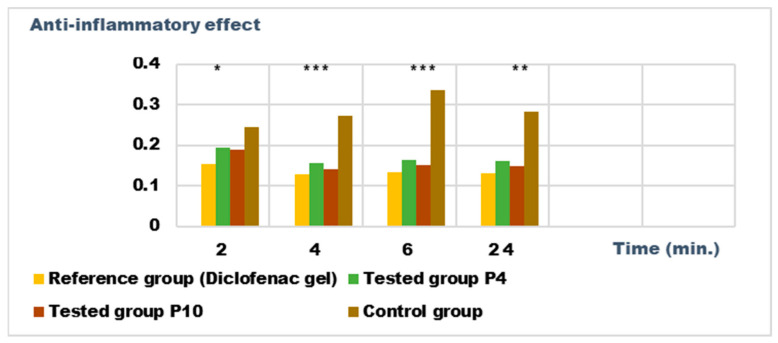
Anti-inflammatory effect on inflammatory edema induced with 10% kaolin suspension. * *p* < 0.05, ** *p* < 0.01, and *** *p* < 0.001 versus the control group.

**Figure 13 ijms-24-16234-f013:**
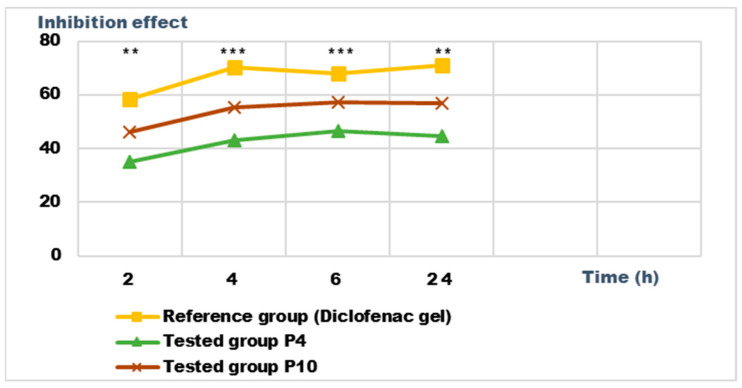
Inhibition effect (I%) on inflammatory edema induced with 10% kaolin suspension during the treatment. *** *p* < 0.001 and ** *p* < 0.01 refer to statistical significance between the reference group and the tested group.

**Figure 14 ijms-24-16234-f014:**
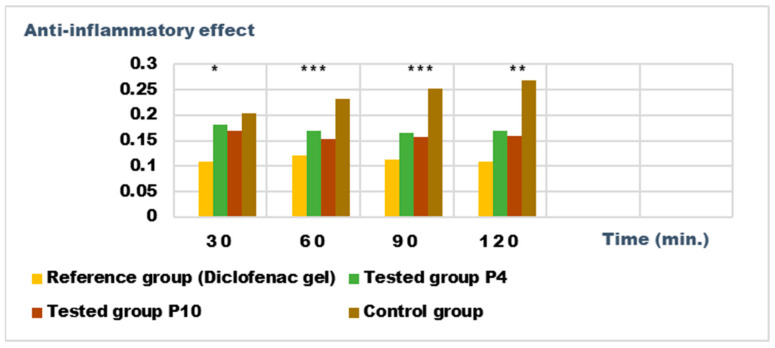
Anti-inflammatory effect on inflammatory edema induced with a 6% dextran solution. * *p* < 0.05, ** *p* < 0.01, and *** *p* < 0.001 versus the control group.

**Figure 15 ijms-24-16234-f015:**
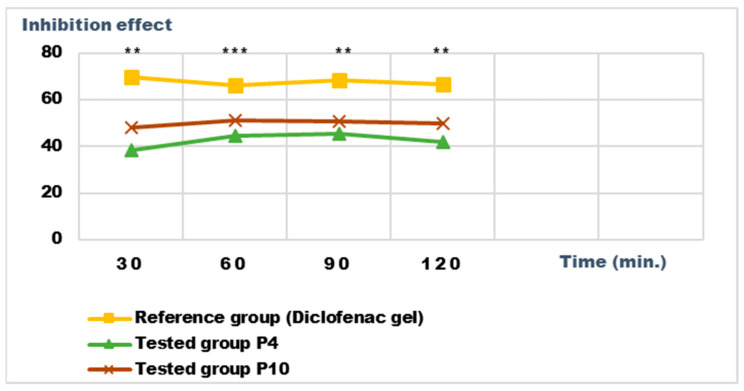
Inhibition effect (I%) on inflammatory edema induced with a 6% dextran solution during the treatment. ** *p* < 0.01 and *** *p* < 0.001 refer to statistical significance between the reference group and the tested group.

**Figure 16 ijms-24-16234-f016:**
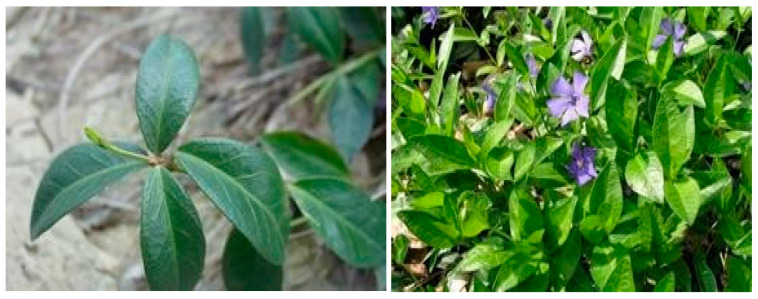
*Vinca minor* L. collected from Siutghiol Lake in the Ovidiu area (44°16′12″ N; 28°33′36″ E), Romania.

**Table 1 ijms-24-16234-t001:** *Vinca minor* stems extract-based biocomposites’ characteristics.

Characteristic	P1	P2	P3	P4	P5	P6
Initial macroscopic characteristics	Homogeneous product; greenish color; specific smell	Homogeneous product; creamy white color; specific smell	Homogeneous product; creamy white color; specific smell	Homogeneous product; white color; specific smell	Homogeneous product; greenish color; specific smell	Homogeneous product; creamy white color; specific smell
Macroscopic characteristics after 30 days	In the initial form	In the initial form	In the initial form	In the initial form	In the initial form	In the initial form
Initial pH	5.8	6.7	5.8	6.7	6.2	6.2
pH after 30 days	5.5	6.5	5.6	6.6	5.8	5.9
Initial thermal stability	Stable, homogeneous product	Stable, homogeneous product	Stable, homogeneous product	Stable, homogeneous product	Stable, homogeneous product	Stable, homogeneous product
Thermal stability after 30 days	Stable, homogeneous product	Stable, homogeneous product	Stable, homogeneous product	Stable, homogeneous product	Stable, homogeneous product	Stable, homogeneous product

**Table 2 ijms-24-16234-t002:** *Vinca minor* leaves extract-based biocomposites characteristics.

Characteristic	P7	P8	P9	P10	P11	P12
Initial macroscopic characteristics	Homogeneous product; greenish color; specific smell	Homogeneous product; greenish color; specific smell	Homogeneous product; green color; specific smell	Homogeneous product; greenish color; specific smell	Homogeneous product; green color; specific smell	Homogeneous product; greenish color; specific smell
Macroscopic characteristics after 30 days	In the initial form	In the initial form	In the initial form	In the initial form	In the initial form	In the initial form
Initial pH	5.8	6.7	6.1	5.4	6.4	6.7
pH after 30 days	5.6	6.4	5.8	5.2	6.1	6.5
Initial thermal stability	Stable, homogeneous product	Stable, homogeneous product	Stable, homogeneous product	Stable, homogeneous product	Stable, homogeneous product	Stable, homogeneous product
Thermal stability after 30 days	Stable, homogeneous product	Stable, homogeneous product	Stable, homogeneous product	Stable, homogeneous product	Stable, homogeneous product	Stable, homogeneous product

**Table 3 ijms-24-16234-t003:** Value intervals for rheological parameters obtained on the *Vinca minor* based biocomposites.

Sample	Shear Spead D (s^−1^) the Interval	Viscosity ƞ (cP) the Interval	Shear Stress τ (mPa) the Interval
P1	1.7–68	1493–17,541	28,090–102,400
P2	2.04–68	2150–25,640	51,102–146,200
P3	1.02–17	30,800–105,100	107,202–351,900
P4	2.04–68	2200–33,198	66,500–150,280
P5	3.4–68	609–45,212	41,400–153,720
P6	3.4–68	1300–12,100	1,690,000–146,410,000
P7	1.7–68	1430–47,950	67,666.8–175,100
P8	4.08–20.04	12,800–35,200	141,168–271,150
P9	1.02–68	4498–72,000	72,420–306,000
P10	2.04–34	8400–102,900	193,016–399,600
P11	2.04–68	4600–56,450	105,056–458,800
P12	3.4–68	4600–43,335	141,576–312,800

**Table 4 ijms-24-16234-t004:** The coefficient values of Ostwald de Waele rheological model.

Biocomposites/Rheological Model	K Consistency Coefficient	n Flow Coefficient	R Correlation Coefficient Ostwald de Waele
P1	10.406	0.3454	0.9988
P2	10.564	0.292	0.9953
P3	11.344	0.7066	0.9979
P4	10.941	0.2345	0.9993
P5	12.388	0.5687	0.9992
P6	20.787	0.5027	0.9962
P7	11.282	0.1800	0.9962
P8	10.647	0.374	0.9992
P9	11.5312	0.24483	0.9970
P10	11.83	0.4038	0.9972
P11	11.540	0.294	0.9976
P12	11.9	0.1841	0.9976

**Table 5 ijms-24-16234-t005:** Antioxidant activity for *Vinca minor* extract-based biocomposites.

No	Sample/Dilution/Working Volume	Free Radicals Max. Inhibition	Total Antioxidant Capacity (nM TE/μL)	TEAC Quantity Means (mg TE/100 g Sample)
1	P3/stock sol./5 μL;	0.945	−3.304	-
2	P3/dil. with ethanol 1:100/5 μL;	0.443	1.959	9804.795
3	P4/stock sol./5 μL;	0.189	0.388	19.419
4	P9/stock sol./5 μL;	0.894	−3.819	-
5	P9/dil. with ethanol 1:100/5 μL;	0.275	0.692	3463.46
6	P10/stock sol./5 μL;	0.228	0.512	25.625

**Table 6 ijms-24-16234-t006:** Percentage composition of biocomposites containing hydroalcoholic macerates from the stem of the plant *Vinca minor* L. (P1–P6).

Components	P1 ^1^	P2 ^1^	P3 ^1^	P4 ^1^	P5 ^1^	P6 ^1^
Distilled water	10.3%	20.5%	26.5%	21.2%	26.8%	27.1%
Hydroalcoholic macerate 40%	-	-	-	-	25%	21.3%
Hydroalcoholic macerate 70%	-	-	24.8%	24.8%	-	-
Hydroalcoholic macerate 96%	16.2%	16%	-	-	-	-
Citric acid	2.6%	-	-	-	-	-
Hyaluronic acid	1%	-	-	-	-	-
Lanolin	62.1%	36%	37%	37%	37.5%	27.1%
Ag sulfadiazine	2.6%	5.1%	5.3%	5.3%	-	6.8%
Cetyl alcohol	2.6%	3%	3.7%	3.7%	10.7%	4.1%
Vaseline	-	10.3%	-	-	-	-
Shea butter	-	5.1%	-	-	-	-
Vitamin E	-	1%	-	-	-	-
Vegetable collagen	2.6%	-	-	-	-	-
Collagen from Nisetru fish	-	-	-	-	-	6.8%
Stearic acid	-	-	2.7%	2.7%	-	-
Zinc oxide	-	3%	-	5.3%	-	6.8%

^1^ Ingredients for 100 g (%).

**Table 7 ijms-24-16234-t007:** Percentage composition of biocomposites containing hydroalcoholic macerate from the leaf of the plant *Vinca minor* L. (P7–P12).

Components	P7 ^1^	P8 ^1^	P9 ^1^	P10 ^1^	P11 ^1^	P12 ^1^
Distilled water	18.7%	20%	26.5%	21.2%	27.6%	26.8%
Hydroalcoholic macerate 40%	-	-	-	-	17.3%	21%
Hydroalcoholic macerate 70%	-	-	24.8%	24.8%	-	-
Hydroalcoholic macerate 96%	14.8%	15.6%	-	-	-	-
Citric acid	2.3%	-	-	-	-	-
Hyaluronic acid	1%	-	-	-	-	1.3%
Lanolin	56%	35%	37%	37%	44.1%	26.8%
Ag sulfadiazine	2.7%	5%	5.3%	5.3%	-	6.7%
Cetyl alcohol	2.3%	3.4%	3.7%	3.7%	11%	4%
Vaseline	-	10%	-	-	-	-
Shea butter	-	5%	-	-	-	-
Vitamin E	-	1%	-	-	-	-
Vegetable collagen	2.2%	-	-	-	-	-
Collagen from Nisetru fish	-	-	-	-	-	6.7%
Stearic acid	-	-	2.7%	2.7%	-	-
Zinc oxide	-	5%	-	5.3%	-	6.7%

^1^ Ingredients for 100 g (%).

**Table 8 ijms-24-16234-t008:** Working scheme of ACL procedure.

Reagents Kit	R1 (μL)	R2 (μL)	R3 (μL)	R4 (μL)	Sample (μL)
Blank	2300 µL	200 µL	25 µL	0 µL	0 µL
Calibration curve	2300 µL	200 µL	25 µL	5 µL	0 µL
Measurement Sample	2300 µL	200 µL	25 µL	0 µL	5 µL

## Data Availability

There are no data available for this publication.

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
