# Peer review of "Development of New Dermato-Cosmetic Therapeutic Formulas with Extracts of Vinca minor L. Plants from the Dobrogea Region"

_ijms, 2023, doi:10.3390/ijms242216234_

Round 1

Reviewer 1 Report

Comments and Suggestions for Authors

I have a few points, mainly regarding the addition of references and the antimicrobial assays. In detail:

Introduction

Line 51: What do you mean by „most characteristic“? Among the Apocyanceae? Does it have more characteristic properties than other plants?

Line 61-63: Please give 1 or more references concerning the use of indole alkaloids in topical formulations.

Line 75-77: PLease add a reference to cite an example for the successful treatment of resistant bacteria through topical applications.

Line 88-103: This paragraph already presents results and should be taken out of the introduction.

 Results

Figures 1 and 2 are very crowded and it’s difficult to read the data. Please modify the Figures to make the data more visible.

Figure 4: The legend overlaps with the figure.

Chapter 2.4 and 2.5 should be combined.

2.4 and 2.5: If the hydroalcoholic macerate was used as “blank” in the sense of “negative control”, this would mean that you expect no activity exerted from the extract. Is that true? Then my question would be why to add the macerate at all. The values for the plant extract should also be added to table 6, otherwise it’s very hard to estimate which activity is caused by the formulation and which by the plant constituents. Or does refering to [20] mean that the antimicrobial effect of the extract has already been tested? Then please add some values about its activity.

In addition, has the ethanol 70 % been taken into account? I.e., were antimicrobial assays initially also performed using only the equivalent volume of extraction solvent as negative control?

Figures 11-13 should be labelled in English.

Figures 14-16: It’s mentioned in the text, but please also add the information what was used as reference (Diclofenac) and as control to the figure legends.

Lines 279-281_ Please also add references here concerning the effects of indole alkaloids.

Lines 319-321: Please add references concerning the phytochemical composition of leaves and stem, respectively, if you make the statement that the extracts have a different composition.

Line 344: Please add references for “biocompatible, biodegradable, non-toxic properties”

 Materials and Methods

Line 365: The statement is very general. If necessary and relevant for the analysis, please give the degree of purity for the respective solvent or chemical. If not, please leave it out.

Lines 373 and 378: Please give a reference where the respective methods are described in detail. Some of them are given later in the text, but I would suggest to describe the methods used only once.

Line 388 / 390: should be „bacterium“

Line 389-393: Please add a reference regarding the information on E.coli, similar to the paragraphs about Staphylococcus and Candida.

Line 398-404: Please doublecheck if this information is correct. Usually, the cell wall of gram-negative bacteria is less permeable than that of gram-positive, and thus, these microorganisms are more resistant to treatment than the gram-positive ones.

I think, the paragraph describing the antimicrobial activity is too detailed. Some of the information could be given in the results / discussion part. Also, there is a separate section (4.3.4) where this information would fit better.

4.2: Please delete „extractive“ from the title

Line 427: Better „ethanol“ instead of „ethyl alcohol“

Line 558: It should probably be „108 CFU“?

Line 622: 11 mm is mentioned here, but not the other diameters. Doesn’t the text refer to all three?

Anti-inflammatory assay: Is there any evidence available from in vitro assays using the respective plant extracts?

Comments on the Quality of English Language

As far as I can judge, the language is fine in general, except for axis labels in Figures 11-13.

Author Response

To reviewer 1

Dear reviewer,

Thank you for the comments made on the manuscript that helped us to improve its quality.

The article has been modified according to the indications received.

Comments and Suggestions for Authors

I have a few points, mainly regarding the addition of references and the antimicrobial assays. In detail:

Introduction

Line 51: What do you mean by „most characteristic“? Among the Apocyanceae? Does it have more characteristic properties than other plants?

            Vinca minor L. is the most characteristic plant from a therapeutic point of view of the Apocinaceae family [1,2], this family includes a large variety of plants. There are about seven species of Vinca (Apocynaceae) in the world. Plants from this family have many uses. Some species are used for their therapeutic effects (Vinca minor L., Catharanthus roseus L.) and some species are cultivated for ornamental purposes (Plumeria rubra L., Carissa L.). This family also includes plants that are considered poisonous (Nerium oleander L.) [3]. In folk medicine, Vinca minor is known for its sedative, hypotensive, antidiabetic effect and also for treating circulatory disorders or promoting cerebral metabolism [2].

  1. Karpus, I.P. Morpho-anatomical studies on Vinca minor L. from the family Apocynaceae. Farm Zh, 1961, 16 (6), p. 48–52.
  2. Panneerselvam, C.; Murugan, K.; Kovendan, K.; Kumar, P.M.; Ponarulselvam, S.; Amerasan, D.; Subramaniam, J.; Hwang, J.S. Larvicidal efficacy of Catharanthus roseus Linn. leaf extract and bacterial insecticide Bacillus thuringiensis against Anopheles stephensi Liston. Asian Pacific Journal of Tropical Medicine, 2013, 6 (11), p. 847-853.
  3. Wong S.K., Lim Y.Y., Chan E.W.C. (2013). Botany, uses, phytochemistry and pharmacology of selected Apocynaceae species: A review. Pharmacogn. Commun. 3, 2013, p. 2.

Line 61-63: Please give 1 or more references concerning the use of indole alkaloids in topical formulations.

We have added to the manuscript the bibliographical references indicated by you:

[4]. Hasa, D.; Perissutti, B.; Cepek, C.; Bhardwaj, S.; Carlino, E.; Grassi, M.; Invernizzi, S.; Voinovich, D. Drug salt formation via mechanochemistry: the case study of vincamine. Mol Pharm. 2013, 10(1):211-24. doi: 10.1021/mp300371f.

We have added to the manuscript the bibliographical references indicated by you.

Line 75-77: PLease add a reference to cite an example for the successful treatment of resistant bacteria through topical applications.

We have added to the manuscript the bibliographical references indicated by you:

[17]. Ciorîță, A.; Zăgrean-Tuza, C.; Moț, A.C.; Carpa, R.; Pârvu, M. The Phytochemical Analysis of Vinca L. Species Leaf Extracts Is Correlated with the Antioxidant, Antibacterial, and Antitumor Effects. Molecules. 2021, 26(10):3040. doi: 10.3390/molecules26103040

Line 88-103: This paragraph already presents results and should be taken out of the introduction.

We eliminated from the introduction the paragraph presenting the results obtained.

 Results

Figures 1 and 2 are very crowded and it’s difficult to read the data. Please modify the Figures to make the data more visible.

We changed

The graphs for Figures 1 and 2 are crowded because flow curves and rheograms usually vary over the same range. In the case of the P3 biocomposite curve which is not included in the graph, the flow curve varies outside the range. The same is applicable for the P6 biocomposite rheogram.  In order to be able to compare these curves we considered it necessary that the flow curves and the rheograms be presented in the same graph. We have also modified the clarity of Figures 1 and 2 to your request (We have attached additional rheology material).

Figure 4: The legend overlaps with the figure.

We changed.

Chapter 2.4 and 2.5 should be combined.

2.4 and 2.5: If the hydroalcoholic macerate was used as “blank” in the sense of “negative control”, this would mean that you expect no activity exerted from the extract. Is that true? Then my question would be why to add the macerate at all. The values for the plant extract should also be added to table 6, otherwise it’s very hard to estimate which activity is caused by the formulation and which by the plant constituents. Or does refering to [20] mean that the antimicrobial effect of the extract has already been tested? Then please add some values about its activity.

In addition, has the ethanol 70 % been taken into account? I.e., were antimicrobial assays initially also performed using only the equivalent volume of extraction solvent as negative control?

Sections 2.4 and 2.5 have been combined following your indication. Table 6 has been eliminated and replaced with Figures 8-11. Also the antimicrobial activity has been reinterpreted (Lines 207-223). Reference [20] (currently [24]) indicates that the antioxidant activity of hydroalcoholic macerates obtained from the leaves and stems of the Vinca minor plant has been tested by us in a previously published research.

Figures 11-13 should be labelled in English.

Figures 11-13 have been eliminated and substituted with Figures 8-11 and then reinterpreted in the manuscript.

Figures 14-16: It’s mentioned in the text, but please also add the information what was used as reference (Diclofenac) and as control to the figure legends.

Figures 14-17 (according to the old numbering), in the present manuscript they are numbered Figures 12-15, have been redone according to your instructions.

Lines 279-281_ Please also add references here concerning the effects of indole alkaloids.

We have added to the manuscript the bibliographical references indicated by you:

[25]. Abouzeid, S.; Hijazin, T.; Lewerenz, L.; Hansch, R.; Selmar, D. The genuine localization of indole alkaloids in Vinca minor and Catharanthus roseus. Phytochemistry 2019, 168, p. 110-112.

[26]. Boga, M.; Kolak, U.; Topcu, G.; Bahadori, F.; Kartal, M.; Farnsworth, N.R. Two new indole alkaloids from Vinca herbacea L. Phytochem. Lett. 2011, 4, p. 399–403.

[27]. Boyadzhiev, L.; Yordanov, B. Pertraction of indole alkaloids from Vinca minor L. Sep. Sci. Technol. 2004, 39, p. 1321–1329.

Lines 319-321: Please add references concerning the phytochemical composition of leaves and stem, respectively, if you make the statement that the extracts have a different composition.

We have added to the manuscript the bibliographical references indicated by you:

[20]. Neculai A.M.; Stanciu G.; Mititelu M. Determination of Active Ingredients, Mineral Composition and Antioxidant Properties of Hydroalcoholic Macerates of Vinca minor L. Plant from the Dobrogea Area. Molecules 2023, 28, p. 5667. https://doi.org/10.3390/molecules28155667

Line 344: Please add references for “biocompatible, biodegradable, non-toxic properties”

We have added to the manuscript the bibliographical references indicated by you:

[28]. Venkatesham, M., Ayodhya, D., Madhusudhan, A., Veera Babu, N., Veerabhadram, G. A novel green one-step synthesis of silver nanoparticles using chitosan: catalytic activity and antimicrobial studies, Appl Nanosci., 2014, 4, p. 113–119.

Materials and Methods

Line 365: The statement is very general. If necessary and relevant for the analysis, please give the degree of purity for the respective solvent or chemical. If not, please leave it out.

We removed this sentence in the manuscript.

Lines 373 and 378: Please give a reference where the respective methods are described in detail. Some of them are given later in the text, but I would suggest to describe the methods used only once.

We have added to the manuscript the bibliographical references indicated by you:

[35]. Popovici, I.; Lupuleasa D. Tehnologie Farmaceutica, 2nd ed; Poliron Iasi, Romania, Vol. 3 (2). ISBN: 9789734669103, 2017, p. 580.

Line 388 / 390: should be „bacterium“

We have done the modification in the manuscript.

Line 389-393: Please add a reference regarding the information on E.coli, similar to the paragraphs about Staphylococcus and Candida.

We have added to the manuscript the bibliographical references indicated by you:

[29]. Dale, A.P., Woodford, N. Extra-intestinal pathogenic Escherichia coli: disease, carriage and clones. J Infect, 201571, p. 615–626.

Line 398-404: Please doublecheck if this information is correct. Usually, the cell wall of gram-negative bacteria is less permeable than that of gram-positive, and thus, these microorganisms are more resistant to treatment than the gram-positive ones.

I think, the paragraph describing the antimicrobial activity is too detailed. Some of the information could be given in the results / discussion part. Also, there is a separate section (4.3.4) where this information would fit better.

We have checked and corrected the paragraph: Line 398-404 (in the present manuscript: line 401-414) and moved it to the section you indicated (4.3.4).

4.2: Please delete „extractive“ from the title.

We have done the modification.

Line 427: Better „ethanol“ instead of „ethyl alcohol“

We have done the modification.

Line 558: It should probably be „108 CFU“?

We have done the modification.

Line 622: 11 mm is mentioned here, but not the other diameters. Doesn’t the text refer to all three?

We have done the explanation in the manuscript (Line 229-235).

Anti-inflammatory assay: Is there any evidence available from in vitro assays using the respective plant extracts?

Some of the phytocompounds present in Vinca minor are also recognized for their anti-inflammatory potential, evident in other plant species where they are found in high concentrations (polyphenols for example), as a result we considered that we can try to investigate the anti-inflammatory effect as well.

Recent studies have highlighted the antioxidant, antibacterial and antitumor effect of the different extracts obtained from the different parts of some Vinca species, effects due to the phytonutrients in the composition of the plants [17]. We set out to analyze the anti-inflammatory potential of the biocomposites made with the extracts obtained from Vinca minor, and as a benchmark we used a pharmaceutical preparation for external use, authorized and recognized for a certain anti-inflammatory effect, with diclofenac as the active substance.

According to the experimental data, a reduction of inflammations slightly over 50% is observed in the case of the P10 formula, which entitles us to say that this formula has a moderate anti-inflammatory potential, which cannot be neglected and which brings a plus in the therapeutic use of the preparation.

Thank you in advance for your understanding,

 Once again, thanks for your help,

The authors

Reviewer 2 Report

Comments and Suggestions for Authors

The manuscript entitled "Development of new dermato-cosmetic therapeutic formulas with extracts of Vinca minor L. plant from Dobrogea" by Ana-Maria Neculai and co-authors presents the study on the potential use of Vinca minor leaves and shoots for preparation of skin regeneration formulas with antimicrobial properties. The study is novel and interesting, however, in the present form the manuscript suffers significant drawbacks that should be revised before resubmission.

1. Title: give plural to “plant” and add “region” at the end.

2. Subsection 2.2. – move to Materials and Methods.

3. Provide statistical analysis for all numeric data presented in charts and tables.

4. Charts are in general of low quality, please correct and unify them.

5. Table 5 – unify decimal places and replace comma with dot.

6. Table 6 – provide statistical analysis in table, not as separate tables (L209: p<0.05 means significance of the effect!)

7. Figs 8-10 – remove and add statistical analysis to previous.

8. Figs 11-13: provide statistics, change language to English.

9. Discussion doesn’t refer to any literature. It must be rewritten in total, in the present form it’s a description of the obtained results not a discussion. If there are not many research on Vinca minor, discuss with other plants to compare.

10.   Materials and Methods: L388-409 – partly move to the Introduction section.

11.   Provide the total dry mass of leaves and shoots taken from the field for preparations.

12.   Table 8 – the percentage is given by mass of by volume? Please give details.

13.   Conclusions are too long and repeat the results. Please rewrite.

Comments on the Quality of English Language

none

Author Response

To reviewer 2

Dear reviewer,

Thank you for your comments on the manuscript that helped us improve its quality.

  1. Title: give plural to “plant” and add “region” at the end.

We change.

Development of New Dermato-cosmetic Therapeutic Formulas with Extracts of Vinca minor L. plants from Dobrogea region

  1. Subsection 2.2. – move to Materials and Methods.

We corrected the typing errors. Subsection 2.2 has been moved to the section you indicated.

  1. Provide statistical analysis for all numeric data presented in charts and tables.

We have done the modification in the manuscript. Also charts have been modified and reinterpreted (Figures 8-11).

  1. Charts are in general of low quality, please correct and unify them.

We changed.

  1. Table 5 – unify decimal places and replace comma with dot.

We have done the modification.

  1. Table 6 – provide statistical analysis in table, not as separate tables (L209: p<0.05 means significance of the effect!)
  2. Figs 8-10 – remove and add statistical analysis to previous.
  3. Figs 11-13: provide statistics, change language to English.

Table 6 and Figures 8-13 have been eliminated and substituted with Figures 8-11 respectively and reinterpreted in the manuscript.

  1. Discussion doesn’t refer to any literature. It must be rewritten in total, in the present form it’s a description of the obtained results not a discussion. If there are not many research on Vinca minor, discuss with other plants to compare.

The Discussion section has also been rewritten according to the instructions received.

  1. Materials and Methods: L388-409 – partly move to the Introduction section.

Lines 388-409 have been moved as indicated in the introduction section (in this manuscript lines 106-117.

  1. Provide the total dry mass of leaves and shoots taken from the field for preparations.

For the preparation of each solution, 50 g of plant product (leaves or stems) from the Vinca minor plant, dried and crushed, were used, over which ethyl alcohol was added in a concentration of 40%, 70%, 96% up to 500 ml respectively (ratio 1:10).

  1. Table 8 – the percentage is given by mass of by volume? Please give details.

We added: Ingredients for 100 g (%)

The percentages are given by the mass of the substances because the preparation process was performed by weighting. We apologise for the typing error in the legend of Tables 6 and 7. A correction has been made in the manuscript.

  1. Conclusions are too long and repeat the results. Please rewrite.

We rewrite:

New pharmaceutical formulations were developed (total of 12), in the form of H/L or L/H emulsions, using hydroalcoholic macerates of concentrations: 40%, 70% and 96%. from the leaves and stems of the Vinca minor plant. To enhance the healing and antimicro-bial effect of the hydroalcoholic macerate from the dermal formulations, substances with their own pharmacological action were used: zinc oxide for its antifungal and antibacteri-al effect, Ag sulfadiazine for its antiseptic effect, hyaluronic acid used for tissue regenera-tion.

The obtained results recommend the use of biocomposites, obtained from indolic het-erocyclics extracted from the Vinca minor plant from the Dobrogea area, in the therapy of skin lesions. Formulations P3 and P9, which show good antioxidant capacity, can also be successfully applied in anti-aging therapy. Formulas P4 and P10 with more pronounced antimicrobial action are recommended in the pharmaceutical industry as an alternative natural preservative and disinfectant. The chemical composition of Vinca minor is corre-lated with the observed antimicrobial potential. Antimicrobial potential of tested compo-sites P3, P4, P9 and P10 was microbial species‐ and dose‐ dependent. Of course, the good antimicrobial effect was due to the synergistic action of the plethora of phytoconstituents, and this was reflected in the antimicrobial activity against bacterial (Gram positive and Gram negative) and fungal strains, but the association of the 70% ethanol extracted bioactive compounds with various substances having their own well known pharmacological action: zinc oxide for its antifungal and antibacterial effect, Ag sulfadiazine for its antiseptic effect, enhance antimicrobial activity by a synergistic effect.

The results of the study confirm that the proposed pharmaceutical formulations, based on hydroalcoholic macerates from the Vinca minor plant, can represent an alternative to classic antibiotics in the development of new topical pharmaceutical preparations with disinfectant activity.

Thank you in advance for your understanding,

 Once again, thanks for your help,

The authors

Round 2

Reviewer 1 Report

Comments and Suggestions for Authors

The revised manscript has well improved. I only have two minor comments:

Line 75-77: This must have been a misunderstanding. What I meant was, please add a reference stating the advantage of semi-solid formulations (unlike other formulations) against resistant bacteria. It was not necessarily related to Vinca minor.

Line 710/711: In the new paragraph it says „use of biocomposites, obtained from indolic heterocyclics extracted from the Vinca minor plant“.

By writing „Vinca minor plant“, you would probably want to indicate the plant part that was used? In this case, I would better write „extracts from the aerial parts of Vinca minor“. 

From previous studies you cited, it’s known that not only indole alkaloids but also different phenolic compounds are present in the plant extract. I would therefore shorten the sentence, for example, as follows: „The obtained results support the use of extracts from the aerial parts of Vinca minor from the Dobrogea area, in the therapy of skin lesions.“

Author Response

Dear reviewer,

Thanks again for your suggestions, which were really helpful in improving the quality of our manuscript.

Regarding the new requirements:

  1. Line 75-77: We have added another bibliographic reference in the manuscript at your indication.

[18]. Brown, T.L.; Petrovski, S.; Chan, H.T.; Angove, M.J.; Tucci, J. Semi-Solid and Solid Dosage Forms for the Delivery of Phage Therapy to Epithelia. Pharmaceuticals (Basel). 2018, 11(1):26. doi: 10.3390/ph11010026.

  1. Line 710/711: We replaced the sentence with the one you indicated.

The obtained results support the use of extracts from the aerial parts of Vinca minor from the Dobrogea area, in the therapy of skin lesions.

Thank you again for your support in improving the quality of the article.

The authors

Reviewer 2 Report

Comments and Suggestions for Authors

The manuscript has been greatly adjusted according to the comments and in my opinion can be accepted for publication.

Author Response

Dear reviewer,

Thank you for your work in reviewing our manuscript, for your suggestions, and for your feedback.

The authors